# Influence of women's legal status on pregnancy outcomes and quality of care: Findings from the Pregnancy of Migrants in Switzerland (PROMISES) program

Eugénie de Weck[1], Clara Noble[1], Jessica Sormani[2], Monique Lamuela Naulin[3], Cyril Jaksic[4], Sara Arsever[5], Begoña Martinez de Tejada[1,3], Nicole C. Schmidt[3,6], Anya Levy Guyer[7], Anne-Caroline Benski[1,3]*

1 Faculty of Medicine, University of Geneva, Geneva, Switzerland, 2 Geneva School of Health Sciences, University of Applied Sciences and Arts Western Switzerland (HES-SO), Geneva, Switzerland, 3 Department of Women, Child and Adolescent, University Hospitals of Geneva, Geneva, Switzerland, 4 Clinical Epidemiology Department, University Hospitals of Geneva, Geneva, Switzerland, 5 Primary Care Division, University Hospitals of Geneva, Geneva, Switzerland, 6 Catholic University of Applied Science, Munich, Germany, 7 Independent Consultant, Boston, Massachusetts, United States of America

* anne-caroline.benski@hcuge.ch

## Abstract

In 2020, approximately 281 million people lived in a country other than their country of birth. In Geneva, Switzerland, people born in other countries constitute 40.2% of the population. We aimed to describe the population of pregnant women delivering at the University Hospitals of Geneva (HUG) maternity department and to identify associations between legal status, migration status, and economic precarity with quality care and health outcomes. We performed an exploratory cross-sectional retrospective study including all women who delivered in HUG's maternity department in May 2019 (n=339). The group was sub-divided in three ways: by migration status (Swiss (S) or migrant (M)); by legal status (documented (D), undocumented (U), or asylum seeker (AS)); and economic precariousness (precarious (P) or non-precarious (NP)). The analysis compared the quality of care received and health outcomes across six sub-groups: Swiss non-precarious women (SNP, 25.3%), Swiss precarious women (SP, 12.2%), documented migrant non-precarious women (DMNP, 34.8%), documented migrant precarious women (DMP, 23.3%), undocumented migrants (UM, 2.4%), and asylum seekers (AS, 2.0%). Precarious patients represented 35.5% of 339 women. Economic precarity was more highly associated with poor health outcomes than legal or migration status. Due to the small samples of undocumented migrants (UM) and asylum seekers (AS), the study had limited capacity to achieve statistical significance for findings. The findings from this exploratory study suggest that, where a program exists to reach pregnant undocumented migrants, a pregnant woman's economic status is also very important in determining her experience with the healthcare system during pregnancy and delivery. More than one third of pregnant women delivering at HUG are in a situation of vulnerability, whether economic or legal. This study had few statistically significant results due to small sample sizes. However, it clearly highlights the need for further research into how best to address various vulnerability factors during pregnancy.

**Data availability statement:** All data described in the manuscript are uploaded as Supporting Information files.

**Funding:** This work was funded exclusively by Fondation Privée des Hôpitaux Universitaires de Genève (https://www.fondationhug.org/) (Grant Number QS09-41 to ACB). The funders had no role in study design, data collection and analysis, decision to publish, or preparation of the manuscript.

**Competing interests:** The authors have declared that no competing interests exist.

## Background

An estimated 281 million people were living in a country other than their country of birth in 2020 and almost one-third (87 million) of the world's international migrants were living in Europe [1]. Among these, 48% were women, almost half of whom were of reproductive age [2]. People migrate to other countries for diverse reasons, including economic and educational opportunities, family connections, or fleeing natural disaster or conflict [1].

In 2020 in Switzerland, 38% of the population was born in another country [3]. This number is equally divided between males and females, putting Switzerland in line with data for Europe in general, in which 51.6% of migrants are women [2]. In Geneva canton, the statistical office (Office cantonal de la statistique, OCSTAT) reported that 60.9% of the population had foreign origins [4].

The migrant population in Switzerland includes people with and without valid residency permits. "Documented" migrants include all persons of foreign origin living in the country on a legal basis (working, studying, etc.) and asylum seekers. Asylum seekers are settled by the Swiss government throughout the country. By the end of 2020, there were 377 asylum seekers in Geneva [5]. "Undocumented" migrants are those without valid residency permits; these migrants without a legal basis for being in the country are also known as "sans-papiers" ("without papers" in English). The sans-papiers either arrive in Switzerland with a tourist visa and then do not leave or enter without a legal permit. An estimated 2.7% of the total population of Geneva—approximately 13,000 people—is undocumented [6].

Migrants in Switzerland, as elsewhere, may experience social, economic, health, and other forms of "vulnerability" [7]. Vulnerable persons are not intrinsically fragile, but they are less protected by collective social structures [8]. Economic insecurity in particular, also known as "precarity", is known to have an impact on health outcomes, in part because it is linked with delayed or no access to healthcare [8].

One source of vulnerability for migrants relates to access to health coverage. In Switzerland, the federal law on health insurance (L'assurance maladie obligatoire, or LAMal) mandates that every person living in the country for more than three months must have health insurance. Asylum seekers in Geneva receive health insurance with the broker Swiss Risk & Care [9]. However, undocumented migrants may not be able to purchase health insurance. Recent data suggest that 90% of undocumented migrants in Geneva canton do not possess any form of health insurance (representing 5-8% of city residents and 2-3% of the canton population) [10].

In addition to the risks inherent in lacking legal status, health coverage, and social security, migrants' individual characteristics can also contribute to vulnerability. Health literacy is known to be a major obstacle for migrant women during pregnancy. There is strong evidence that vulnerability among migrant women affects their access to health care and support during pregnancy. Access to antenatal care for undocumented migrants has been described as sporadic [11]. Being a refugee or migrant is considered a risk factor for limited or delayed access to health care; migrant status may also be associated with other risk factors, including lower socioeconomic status, higher burden of disease in the country of origin, or language difficulties [12–14]. Vulnerable pregnant migrants have higher rates of adverse obstetrical and perinatal outcomes [13], including higher rates of mortality and morbidity [13,15,16] and more frequent mental health issues [17,18]. Delays in accessing care are frequent, including postponing antenatal care to prioritize more urgent situations [12]. Lack of patient follow-up after delivery is another major issue for migrant women, who are susceptible to chronic illnesses and other health problems [19–21].

Studies investigating social determinants that affect the health of pregnant migrants in Switzerland have identified several factors, including: financial constraints, language barriers, perceived or documented discrimination, feelings of embarrassment during obstetric visits, stress due to legal status, and lack of family support networks [22,23]. Delayed and sporadic patterns of access to antenatal care have also been recorded in Geneva's undocumented women [24].

The number of migrant women of childbearing age living in Switzerland is growing [25]. Responding to the diverse health care needs of pregnant women with migrant backgrounds presents a complex challenge [22,26]. WHO technical guidance on the health care needs of migrant women recommends: increasing migrants' and health professionals' awareness of programs that address migrant patients' needs, expanding access to translation services, and addressing affordability [27].

The University Hospitals of Geneva (HUG) is the leading health care provider in the Geneva canton. The HUG maternity department is the only public maternity hospital in the canton. In 2019, 83% of births (4248 out of 5111) in the Geneva canton occurred there [28,29]. HUG has already adapted its services to better fit vulnerable migrant women. The CAMSCO program (Ambulatory Mobile Community Care Consultation) was established in 1997 to foster access to health care for vulnerable populations. CAMSCO provides pregnant undocumented migrant women with access to health care provided by midwives specialized in migrant health. Furthermore, interpreters are available at HUG. However, in the past decade, no quantitative analyses have been conducted on data from the HUG maternity department to examine access to health care, the quality of health care provided, or obstetrical outcomes among vulnerable migrant pregnant women [30].

Therefore, the main objective of this study was to use existing data to explore potential associations among nationality, legal status, and quality of care with obstetric outcomes. The study began with the hypothesis that legal and migration status are important aspects of vulnerability that affect the quality of care and health outcomes of pregnant women. The underlying objective was to identify characteristics of vulnerable pregnant women in order to inform HUG's ongoing interventions to improve the care provided to this population. A larger study that incorporates three years of data is planned as a follow-up to this exploratory study.

## Methodology

### Ethics Statement

The study was approved on 29 August 2022 by the Cantonal Ethics Board of Geneva, Switzerland (N° 2022-01013). The research project was accepted under Article 34 on the re-use of personal health data in the absence of consent.

### Setting and study design

This exploratory cross-sectional retrospective study utilized existing, routinely collected data on all 339 women who delivered a newborn (alive or dead, at any gestational age) at the HUG maternity department in May 2019.

### Data collection

S1 Table and S2 Table detail the data points that were collected or generated for every woman who delivered at HUG between 1 May and 31 May 2019. Most data points were extracted from HUG's electronic medical records and administrative files in September 2022. Ultrasound records were either drawn from the patient medical records or were manually searched (the latter was mainly the case for undocumented migrant women). The data were

anonymized, coded, and stored in a database on the HUG server. The medical records were incomplete for some variables (notably ultrasound, language proficiency, and intimate partner violence). Some variables (such as education level) used unclear or inconsistent classifications. Because the study anonymized the data, we had no access to information that could identify individual participants and could not trace missing data points. Several "missing values" therefore appear in the tables. In addition to the medical record data, public data extracted from the Swiss Federal Statistical Office (FSO) and OpenStreetMap in January 2023 were used to generate a precarity variable. The ISCO-08[1] structure was used to classify levels of occupation [31].

## Defining vulnerability factors

The women included in the study were divided into four categories of legal status: 1) Swiss citizens, 2) documented migrants, 3) undocumented migrants, and 4) asylum seekers. These four groups correspond to three insurance status categories: Swiss citizens and documented migrants with mandatory LaMal insurance, undocumented migrants with no insurance[2], and asylum seekers covered by Swiss Risk & Care.

Precarity status for Swiss and documented migrants was assigned based on the place of residence, combining geographical and neighborhood economic data points[3]. We defined precarity as living in a sub-sector where the median income was below the canton's 2021 minimum wage (of CHF 50,538 per annum) and then designated each patient as either precarious (P) or non-precarious (NP).

Undocumented migrants and asylum seekers were excluded from the geographical analysis due to an assumption of precarity, but also because their addresses are less predictive of their economic status, as residency is mostly attributed or not selected by choice. Women living outside of Geneva and those living in a sub-sector for which income data was not available were not included in the analyses relating to precarious financial situations.

This ultimately resulted in six comparison groups: 1) Swiss non-precarious (SNP) women; 2) Swiss precarious (SP) women; 3) documented migrant non-precarious (DMNP) women; 4) documented migrant precarious (DMP) women; 5) undocumented migrant (UM) women; and 6) women asylum seekers (AS).

## Defining independent and dependent variables

Migration status, legal status, and precarity were the independent variables in the analysis (S1 Table). The dependent variables (S2 Table) fell into two categories: quality of care and health outcomes. Two quality-of-care variables were chosen based on international or Swiss recommendations [34]: timely ultrasound (US) (defined as undergoing two ultrasounds, one between 10 and 14 weeks and a second between 18 and 24 weeks of gestation)); and appropriate time for first contact (defined as within the first 12 weeks of pregnancy) [35].

---

[1]  The International Standard Classification of Occupation (ISCO-08) determines 8 major groups of occupation with several sub major and minor groups: Managers; Professionals; Technicians and Associate Professionals; Clerical Support Workers; Services and Sales Workers; Skilled Agricultural, Forestry and Fishery Workers; Craft and Related Trades Workers; Stationary Plant and Machine Operators; and Elementary Occupations. This 2007 classification was made by a group of experts on labor statistics [31].
[2]  At HUG, undocumented migrants who do not have insurance can access care freely after a social evaluation. They are then labeled administratively as "non billable patients". This label has been used as a surrogate measure for undocumented migrants.
[3]  Using each patient's residential address from her medical record, we generated GPS coordinates using the Nominatim search engine from the open-source software OpenStreetMap, available under the Open Database License [32]. The GPS coordinates were then linked with one of the 475 statistical sub-sectors in Geneva canton using data provided by the Territory Information System in Geneva (SITG) [33]. Each patient record was assigned to the median income for a single adult living in their sub-sector using Canton of Geneva official statistics.

## Analysis

All analyses were conducted using the statistical software R (version 4.2.2.). Continuous variables are reported as medians and first and third quartiles, and categorical variables are reported as count and percentages. Comparisons between the documented migrant non-precarious (DMNP) and documented migrant precarious (DMP) women were done using Mann-Whitney U tests for continuous variables, and chi-squared or Fisher's exact tests for categorical variables. A threshold of alpha=0.05 was used to determine statistical significance. The study faced a notable limitation due to the small samples of undocumented migrants (UM) and asylum seekers (AS), which restricted its capacity to achieve statistical significance.

## Results

### Study population

As shown in Fig 1, 339 women delivered a newborn (alive or dead, at any gestational age) in the HUG maternity department in May 2019. Swiss patients who were not residents of Geneva canton (n=43) were excluded, leaving 296 patients included in the final analysis. These patients were divided into six groups for the analyses, as shown in Fig 1: Swiss non-precarious (SNP) women (25.3%, n=75), Swiss precarious (SP) women (12.2%, n=36), documented migrant non-precarious (DMNP) women (34.8%, n=103), documented migrant precarious (DMP, n=69) women (23.3%), undocumented migrant (UM) women (2.4%, n=7) and women asylum seekers AS (2.0%, n=6).

### Socio-demographic characteristics of the study population

All demographic data on the study population are included in S3. Fig 2 shows the countries of origin of the women. The majority of both DMNP and DMP women were from Western Europe. UM women came mainly from South America; others migrated from Asia and Africa. AS women migrated mainly from Africa, with some from Asia.

Swiss and documented patients, whether precarious or not, had similar median ages (SNP: 31; SP: 31; DMNP: 32; DMP: 32 years), while the median ages of UMs and ASs were lower (UM: 26; AS: 29 years). Approximately two-thirds of the sample, including more than three-quarters of DMP (75.4%) and AS (83.3%), were delivering their second child. However, among the UMs, more than 50% (4 out of 7) were pregnant for the first time. Parity was highest (2 or more births) in the AS (50.0%), SP (30.6%) and DMP (26.1%).

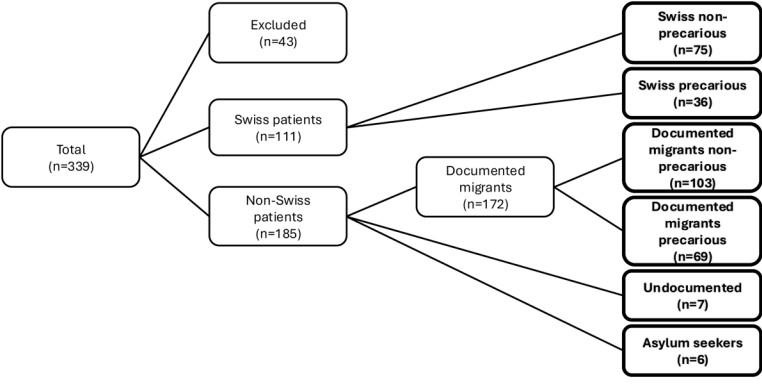

**Fig 1. Distribution of patient groups.**

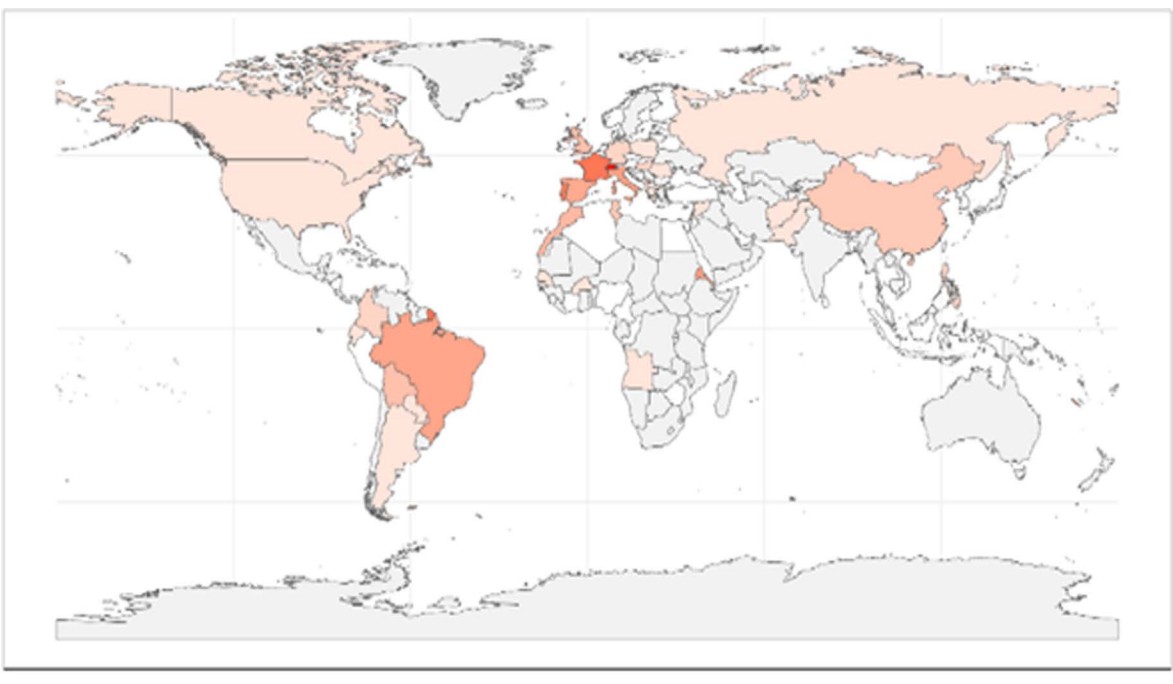

**Fig 2. Nationalities of women delivering at the HUG maternity in May 2019.**

Regarding marital status, the majority (83.3%) of AS, as well as Swiss and documented migrant women were married (SNP 50.7%, SP 58.3%, DMNP 69.9%, DMP 69.6%). Among the UM women, 57% were single.

French was listed as the first language of more than half of the patients in all groups (including six of the seven UMs). However, a quarter (25.7%) of the patients reportedly did not speak French as either their first or second language.

SNP women had the lowest rate of unemployment (13.3%). Most UM women were not employed (57.1%). The six AS women were also not working, as it is prohibited by Swiss law[4]. Among documented migrants, the fraction of women without occupations was higher in the DMP group (46.4%) compared to the DMNP group (28.2%). The association was opposite among women with high skilled work (35.9% of DMNPs, 20.3% of DMPs, p-value = 0.015) (see S3 Table).

All Swiss patients (SNP and SP) were insured by LaMAL, as were the majority (over 90%) of both groups of documented migrants (92.2% of DMNP, 98.6% of DMP). A very small proportion of our sample had international insurance (2.9% of DMNP, 1.4% of DMP) or paid out-of-pocket (4.8% of the DMNP). By definition, the six AS women were covered by the Swiss Risk & Care insurance broker; the seven UMs had no insurance and were considered non-billable patients.

---

[4] In Switzerland, asylum seekers have a "permit N" that does not allow them to work. Documented migrants, including refugees ("permit B"), are allowed to apply for work permits [36].

## Obstetrical outcomes

Complete details on the obstetrical outcomes for the HUG patient population in May 2019 are presented in S4. Outcomes of particular interest include:

**Duration of pregnancy.** The overall preterm delivery rate was 6.1%. SPs had the highest rate (11.1% of deliveries) of preterm births, followed by SNPs (6.7%). The migrant groups had lower rates of preterm deliveries (DMNPs: 4.8%; DMPs: 5.7%; all pregnancies of UMs and ASs were all full-term).

**Pregnancy complications.** Despite what might have been predicted based on the literature, in our sample the ASs had no diagnosed complications (such as gestational diabetes/diabetes, preeclampsia, or gestational hypertension) [37,38]. However, it was noted that 66.7% of asylum seekers had undergone female genital cutting (compared to 4% of DMNPs, 6.2% of DMPs and none of the UMs, SPs and SNPs). This condition can lead to complications during delivery, including increased instrumented labor, C-section or postpartum hemorrhage, among others [39].

**Type of delivery.** All modes of delivery were represented in the six groups. However, vaginal birth was least frequent among Swiss patients regardless of precarity status. None of the AS had C-sections (see Fig 3).

**Labor complications.** Fig 4 shows the prevalence of three maternal complications of labor (type II or III perineal tear, episiotomy, and postpartum hemorrhage) that have strong associations with health outcomes [40]. Episiotomies occurred in 7.8% of cases, with the highest proportion among ASs (33.3%) and none among UM women. Postpartum hemorrhages occurred in 16.2% of women in the sample. In this instance, the UMs had the highest rate (28.6%; 2 out of 7), while no postpartum hemorrhage was recorded among ASs.

## Quality of care

The complete table of quality results are detailed in S5. Notable findings include

**First contact with health system.** The majority of women in the sample had their first pregnancy-related contacts with the health care system through planned consultations (66.7% for the SNPs and SPs, 73.8% of DMNPs, 72.5% of DMPs, 100% of UMs, 83.3% of ASs). The remaining women first accessed health services during their pregnancies via emergency care (see Fig 5).

**Site of routine antenatal care.** A majority of Swiss patients (80% of the SNPs and 91.7% of the SPs) attended private antenatal clinics. This proportion was lower for both groups of documented women (DMNPs: 73.8%, DMPs: 66.7%). Among the women with precarious economic status, significantly fewer DMP women used private gynecological monitoring compared with SP women (SP 91.7%, DMP 66.7%, p-value = 0.010). Private antenatal care was used by only one UM woman and none of the AS women.

**Timely ultrasound (US) and first contact.** Data on "Appropriate time for first contact" and "Timely US" [35] were available for a subset of the sample, namely women who exclusively used HUG's maternity services (i.e., did not use private gynecological clinics) (N=81; SNP n=15, 18.5%; SP n=3, 3.7%; DMNP n=27, 33.3%). Four UM women (66.7%) had an appropriate time for first contact; the other two had their first monitoring visits late, after the first trimester (see S6 Table). One of these two had neither a timely ultrasound nor an appropriate time for first contact. Over 80% of Swiss patients had a timely ultrasound. On the other hand, we found lower rates of timely ultrasound among both AS (66.7%) and UM women (33.3%).

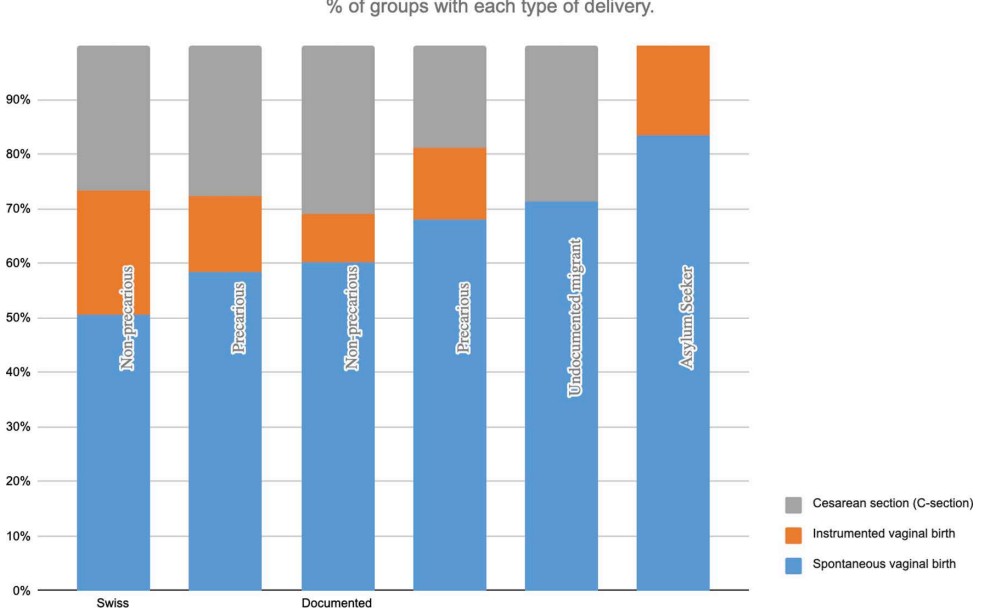

**Fig 3. Percentage of type of delivery in each group.**

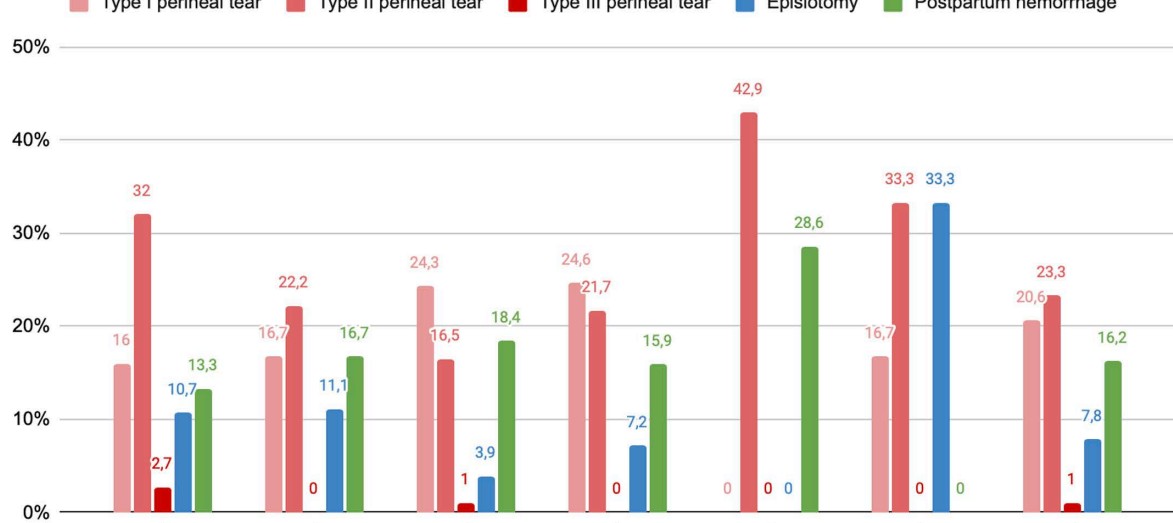

**Fig 4. Labor complications' variables.**

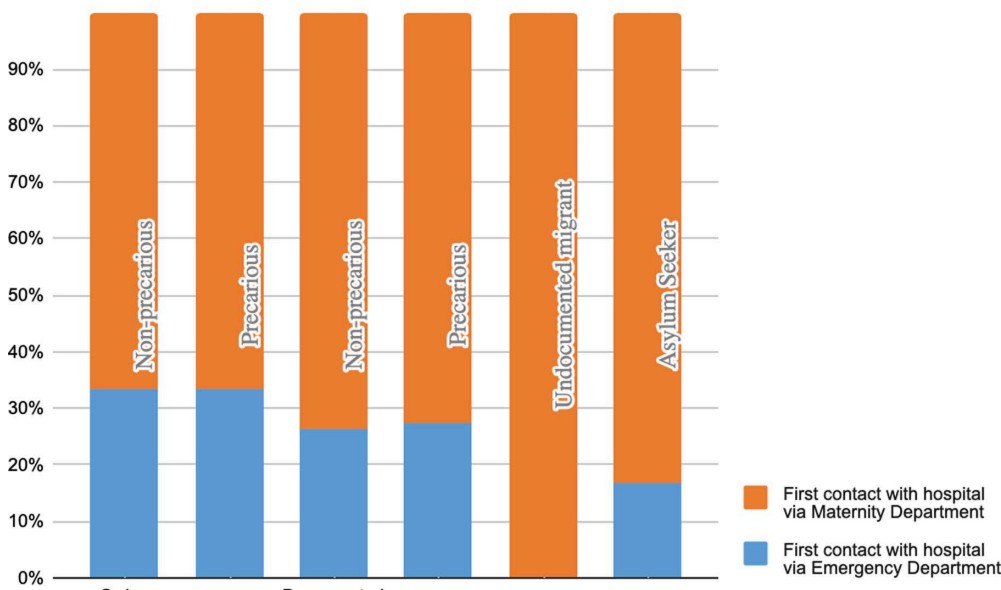

**Fig 5. Entry point of care at the HUG for patients in each group.**

## Discussion

This exploratory study examined the association of three dimensions of vulnerability (either documented or which could be extrapolated from data routinely collected in HUG's electronic medical record system) with key aspects of quality of obstetrical care and obstetrical health outcomes. Some of the results were anticipated while others were more surprising.

### Importance of economic status to health and health care

This study was originally conducted to examine associations between a woman's legal and migrant status and obstetrical care quality and outcomes. Our preliminary hypothesis was that women who were undocumented or seeking asylum would have worse obstetrical outcomes than Swiss citizens and documented migrants related, at least in part, to lower quality of care. However, the study revealed that economic precarity, rather than legal status, had the strongest association with several indicators of poor health outcomes.

A surprisingly high proportion (35.5%) of HUG's Swiss or documented obstetric patients face economic precarity (including 12.2% of SP and 23.3% of DMP women). This is almost four times higher than in Switzerland overall in 2019, when 8.7% of the population was described as poor, and 12.2% as having difficulties providing for their needs [41]. This key finding highlights that a patient's economic status may influence their experience with the Swiss healthcare system and health outcomes [42,43]. The study also found associations between a patient's legal status and other factors related to economic precarity, including employment status, marital status, and nation of origin.

### Meeting the needs of migrant women during pregnancy and delivery

The study's findings related to both obstetrical outcomes and quality-of-care indicators suggest that HUG is generally meeting the obstetrical care needs of migrant patients. For

example, among the obstetrical outcomes examined, we found a lower-than-expected prevalence of labor complications, compared with the latest statistics [44] and no definitive correlation with legal status. Among the patient groups at HUG, those with the highest rates of tearing were UMs (type II: 42.9%, type III: 0.0%), SNPs (type II: 32.0%, type III: 2.7%), and ASs (type II: 33.0%, type III: 0.0%). While lower than expected, these results still suggest that the most vulnerable women are experiencing worse outcomes.

One quality-of-care indicator used in the study is whether a pregnant woman's first contact with the healthcare system during pregnancy is via antenatal care or emergency services. (First healthcare contact during pregnancy via the emergency room can indicate that a woman has no other entry points to health care.) In the study sample, none of the UM women and only one AS woman had first contact via emergency services. The Swiss patients were more likely to use the emergency room for a first contact. This may be partly related to having access to private medical care, in which case women only enter the HUG system for emergency reasons.

The study results suggest that the CAMSCO program seems to be effective in linking undocumented pregnant migrant women to maternity health care in Geneva. This is encouraging, as access to primary care is a positive determinant of follow-up for migrant women in high-income countries [45], and lack of continuity of care has negative effects [46]. However, CAMSCO reaches only approximately 100 women annually. This study's findings on the nature of vulnerability among pregnant women suggest that other programming is required to address the needs of pregnant women facing economic precarity, regardless of their immigration status.

## Limitations

The main limitation of this exploratory study was the sample size, which was not powered to generate statistically significant findings when comparing the sub-groups. While these findings can be used as a baseline for comparison for future studies, it is difficult to use them to justify recommendations for intervention. The findings also raise concerns about the quality and consistency of the data collected in HUG's electronic medical records. If the data are incomplete—or cannot be recorded with appropriate nuance—in the current system, it undermines the usefulness of the system and analyses of the data. For example, while the data indicated that French language was commonly spoken in all patient groups, empirical experience raises concerns about the validity of this finding. Inaccurate data on language skills can result in insufficient for translation services during health care [47]. Another questionable finding was that no intimate partner violence was documented. This finding does not align with previous literature [48] or empirical experience. There are several possible explanations for the discrepancy: the data collection process used when completing the electronic patient file may not be conducive to asking sensitive questions, or healthcare workers may decide not to include the data in the file for privacy or security reasons. Yet screening for intimate partner violence during pregnancy is essential, as it is correlated to poor maternal health outcomes [49].

## Conclusion

This exploratory study started with the hypothesis that legal status is an important aspect of vulnerability that affects both the quality of care provided to pregnant women and their health outcomes. It explored associations among pregnant women's legal, migration, and precarity statuses, as well as other demographic characteristics, and assessed these characteristics' associations with indicators of health care quality and health outcomes.

Other studies have demonstrated stronger links between migration status and clinical outcomes [50]. This study, however, highlighted that when programs exist to

address vulnerability related to women's legal and migration status, economic precarity is a major source of vulnerability. Our findings suggest that HUG is providing quality care to legally vulnerable women, including undocumented women and asylum seekers. This is partly thanks to the CAMSCO program, which coordinates interdisciplinary health interventions for key vulnerable populations in Geneva; it has set precedents for how to integrate socio-economic and cultural support to address social determinants of health [51].

The study also explored other dimensions of vulnerability among the obstetrical patient population at HUG. This led to another key finding: economic precarity, in addition or regardless of legal status, was highly associated with poorer health outcomes and lower quality care for pregnant women.

While these key findings are informative, this was designed as an exploratory study and lacked statistical power. It has provided several important topics for additional research efforts with larger sample sizes and qualitative components. These include further exploration of the relationships among different dimensions of vulnerability, the types of targeted services pregnant women require, and the determinants of health outcomes. Future quantitative studies at HUG should include larger sample sizes to better understand the nature of the associations among demographic characteristics (such as precarity, employment status, language proficiency, and access to private prenatal care) with clinical and quality outcomes. Qualitative research on communication between HUG's migrant patient population and its health care providers could suggest additional ways to improve the quality of care provided to vulnerable patients.

These additional areas of research can further inform improved practices in obstetric care and support at HUG and other hospitals. Potential interventions may include: identifying key variables that should be added to patients' electronic medical records to improve quality of care; additional training for obstetrical staff to improve their understanding and responses to different dimensions of vulnerability; increasing inter-departmental and inter-disciplinary collaboration to more effectively reach and support vulnerable women and their children across the life cycle; and utilization of new technologies, such as artificial intelligence, to mine electronic medical records for more nuanced analyses of both vulnerability and successful approaches to providing quality obstetrical care.

## Supporting information

**S1 Table.** Independent variables.
(DOCX)

**S2 Table.** Demographic and dependent variables.
(DOCX)

**S3 Table.** Sociodemographic variables, disaggregated by group.
(DOCX)

**S4 Table.** Obstetrical variables, disaggregated by group.
(DOCX)

**S5 Table.** Quality variables, disaggregated by group.
(DOCX)

**S6 Table.** Quality variables for women with no private care.
(DOCX)

**S7 Table.** Sociodemographic variables comparing precarious vs. non-precarious documented migrant women.
(DOCX)

**S8 Table.** Obstetrical variables comparing precarious vs. non-precarious documented migrant women.
(DOCX)

**S9 Table.** Quality variables comparing precarious vs. non-precarious documented migrant women.
(DOCX)

**S10 Table.** Sociodemographic variables for precarious women, Swiss vs. documented migrants.
(DOCX)

**S11 Table.** Obstetrical variables for precarious women, Swiss vs. documented migrants.
(DOCX)

**S12 Table.** Quality variables for precarious women, Swiss vs. documented migrants.
(DOCX)

## Author contributions

**Conceptualization:** Eugénie de Weck, Clara Levy Noble, Anne-Caroline Benski.

**Data curation:** Clara Levy Noble, Anne-Caroline Benski.

**Formal analysis:** Eugénie de Weck, Clara Levy Noble, Jessica Sormani, Cyril Jaksic.

**Methodology:** Eugénie de Weck, Clara Levy Noble, Jessica Sormani, Monique Lamuela Naulin, Cyril Jaksic, Sara Arsever, B. Martinez de Tejada, Nicole C. Schmidt, Anya Levy Guyer, Anne-Caroline Benski.

**Supervision:** Anne-Caroline Benski.

**Validation:** Eugénie de Weck, Clara Levy Noble, Jessica Sormani, Monique Lamuela Naulin, Cyril Jaksic, Sara Arsever, B. Martinez de Tejada, Nicole C. Schmidt.

**Writing – original draft:** Eugénie de Weck, Clara Levy Noble, Anne-Caroline Benski.

**Writing – review & editing:** Eugénie de Weck, Clara Levy Noble, Monique Lamuela Naulin, Sara Arsever, B. Martinez de Tejada, Nicole C. Schmidt, Anya Levy Guyer, Anne-Caroline Benski.

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
