## [Decision Letter · Decision Letter 0]

22 Nov 2024

PGPH-D-24-01793

Influence of women’s legal status on pregnancy outcomes and quality of care: Findings from the Pregnancy of Migrants in Switzerland (PROMISES) program

Dear Dr. Benski,

Thank you for submitting your manuscript to PLOS Global Public Health. After careful consideration, we feel that it has merit but does not fully meet PLOS Global Public Health’s publication criteria as it currently stands. Therefore, we invite you to submit a revised version of the manuscript that addresses the points raised during the review process.

We look forward to receiving your revised manuscript.

Kind regards,

Ferdinand C Mukumbang, PhD

Academic Editor

Journal Requirements:

Additional Editor Comments (if provided):

Reviewers' comments:

Reviewer's Responses to Questions

**Comments to the Author**

1. Does this manuscript meet PLOS Global Public Health’s publication criteria ? Is the manuscript technically sound, and do the data support the conclusions? The manuscript must describe methodologically and ethically rigorous research with conclusions that are appropriately drawn based on the data presented.

Reviewer #1: Yes

Reviewer #2: Yes

2. Has the statistical analysis been performed appropriately and rigorously?

Reviewer #1: Yes

Reviewer #2: I don't know

3. Have the authors made all data underlying the findings in their manuscript fully available (please refer to the Data Availability Statement at the start of the manuscript PDF file)?

Reviewer #1: Yes

Reviewer #2: Yes

4. Is the manuscript presented in an intelligible fashion and written in standard English?

Reviewer #1: Yes

Reviewer #2: Yes

5. Review Comments to the Author

Reviewer #1: Thank you for submitting your manuscript. I have reviewed the article and would like to offer some suggestions to improve its quality.

Sample Size in Methods: In the methodology section, the sample size is not clearly stated. I recommend explicitly mentioning the sample size used for the study to provide transparency and help readers assess the robustness of the findings.

References: I noticed that several references are quite old, from 2004 or 2008. To ensure the relevance and currency of the research, please replace these older references with more recent ones from 2010 onwards. This will provide a stronger foundation for your study, reflecting the latest research and developments in the field.

Thank you for considering my feedback. I believe these revisions will enhance the clarity and quality of the manuscript.

Reviewer #2: This paper discusses the influence of women's legal status on pregnancy outcomes and quality of care in Switzerland, specifically focusing on migrants and asylum seekers. The hypothesis was that legal and migration status affects the quality of care and health outcomes of pregnant women - as has been found elsewhere. What was therefore illuminating about this study is that economic precarity was found to have a greater influence on pregnancy outcomes. The finding that economic insecurity is a significant aspect of the vulnerability facing all pregnant women is important, and showcases the way that economic marginalisation is cross-cutting affecting citizens and no-citizens alike. While individual characteristics were also found to have some significance (language barriers, health literacy and access to private antenatal care), the systemic nature of economic precarity make it rich for further investigation.

The study was rigorously performed, well executed and informative.

I have made the suggestion that vulnerability be included as a key word and in the title of the paper.

6. PLOS authors have the option to publish the peer review history of their article (what does this mean? ). If published, this will include your full peer review and any attached files.

**Do you want your identity to be public for this peer review?** For information about this choice, including consent withdrawal, please see our Privacy Policy .

Reviewer #1: No

Reviewer #2: **Yes: ** Nicole Miriam Daniels

---

## [Editor Report · Decision Letter 1]

23 Jan 2025

Influence of women’s legal status on pregnancy outcomes and quality of care: Findings from the Pregnancy of Migrants in Switzerland (PROMISES) program

PGPH-D-24-01793R1

Dear Dr Benski,

We are pleased to inform you that your manuscript 'Influence of women’s legal status on pregnancy outcomes and quality of care: Findings from the Pregnancy of Migrants in Switzerland (PROMISES) program' has been provisionally accepted for publication in PLOS Global Public Health.

Best regards,

Ferdinand C Mukumbang, PhD

Academic Editor